# Sociodemographics and health-literacy as predictors of cervical cancer screening practices among Haitian women: A secondary data analysis of 2016–17 DHS surveys

**Dominique Guillaume** [1,2,3] *, **Rhoda Moise** [4], **Joyline Chepkorir** [1], **Kamila Alexander** [1], **Maria Luisa Alcaide** [5], **Rasheeta Chandler** [6], **Claire Rolland** [7], **Natalie Pierre-Joseph** [8]

1 Center for Infectious Disease and Nursing Innovation, School of Nursing, Johns Hopkins University, Baltimore, MD, United States of America, 2 Jhpiego, A Johns Hopkins University Affiliate, Baltimore, MD, United States of America, 3 International Vaccine Access Center, International Health Department, Johns Hopkins University Bloomberg School of Public Health, Baltimore, MD, United States of America, 4 Center for Translational Sleep and Circadian Sciences, Department of Psychiatry and Behavioral Services, University of Miami Miller School of Medicine, Miami, FL, United States of America, 5 Division of Infectious Diseases, Department of Medicine, University of Miami Miller School of Medicine, Miami, FL, United States of America, 6 Nell Hodgson Woodruff School of Nursing, Emory University, Atlanta, GA, United States of America, 7 Drexel University College of Medicine, Philadelphia, PA, United States of America, 8 Department of Pediatrics and Adolescent Medicine, Boston University School of Medicine, Boston Medical Center, Boston, MA, United States of America

* dguilla2@jhu.edu

**Data Availability Statement:** The data used from this study is made publicly available through the

## Abstract

Cervical cancer screening rates in Haiti are concerningly low. Access to health-related information and health literacy may be important determinants of engagement in cervical cancer screening. This study explored the relationship between sociodemographics, literacy, and sources of information on cervical cancer screening among Haitian women. A secondary data analysis was conducted using USAID Demographics and Health Survey Haiti household data from 2016–2017. Univariate logistic regressions identified significant predictor covariates measuring sociodemographics and sources of information in cervical cancer screening uptake. Two multivariate logistic regression models with adjusted odds ratios were developed using the significant predictor variables from the univariate analysis. N = 610 women responded to questions pertaining to cervical cancer screening. The first multivariate model evaluating sociodemographics demonstrated an economic background of poorer (aOR = 4.06, 95% CI [1.16,14.27]) and richest (aOR = 19.10 , 95% CI[2.58,141.57]), higher education levels (aOR 7.58 , 95% CI [1.64,34.97]), and having insurance (aOR = 16.40, [95% CI 2.65, 101.42]) were significant predictors of cervical cancer screening. The second model evaluating literacy and sources of information indicated that access to a television (aOR = 4.28, 95% CI [1.21,9.34]), mobile phone ownership (aOR = 4.44, 95% CI [1.00,5.59]), and reading the newspaper (aOR = 3.57, [95% CI 1.10,11.59]) were significant predictors of cervical cancer screening. Diverse health communication initiatives that are adapted for literacy level and that incorporate multimedia components may effective in raising women's cervical cancer knowledge and awareness , and increasing intention and uptake of cervical cancer screening in Haiti.

USAID DHS Program: https://dhsprogram.com/Data/ .

**Funding:** This research was supported by the Jhpiego Global Women's Health Fellowship which was obtained by the lead author (DG). The funder had no role in study design, data collection and analysis, decision to publish, or preparation of the manuscript.

**Competing interests:** The authors have declared that no competing interests exist.

## Introduction

Cervical cancer, which is primarily caused by infection with human papillomavirus (HPV), is the fourth most diagnosed cancer amongst women globally [1]. Haiti reports the highest rates of cervical cancer incidence and mortality in the Caribbean, with cervical cancer being the second most diagnosed female cancer among women between 15–44 years old [2–4]. The mortality associated with cervical cancer in Haiti is alarming- amongst those who are diagnosed, approximately 75% will die from the disease [4]. While data on the prevalence of HPV in Haiti has been inconsistent, certain studies have estimated prevalence rates of high-risk oncogenic HPV strains in Haiti being as high as 19% in semi-urban and urban populations [5]. Due to the lack of cancer registries and insufficient cancer surveillance systems in Haiti, it is likely that these numbers are higher than what is reported.

Although cervical cancer can be averted through primary prevention (i.e. HPV vaccination) and secondary prevention (i.e. cervical cancer screening), women in Haiti are less likely to engage in preventative measures [3, 6]. To date , HPV vaccines are not included into Haiti's national immunization schedule. Therefore, the main prevention method that is utilized is cervical cancer screening. The success of cancer screening programs is primarily dependent on the support of the local health system, coupled with patient engagement and adherence [7, 8]. Currently, the standard of care for cervical cancer screening by the Haitian Ministry of Health (Ministre de la Sante Publique et de la Population [MSPP]) is cytology with Papanicolaou tests (i.e. pap smears) [6]. However, numerous challenges exist in implementation due to insufficiently trained staff to conduct pap smears, lack of labs and pathologists to interpret results, and delayed notification of positive results [9, 10]. These limitations are not only unique to Haiti, but to many low- and -middle-income countries (LMICs), which has resulted in the WHO establishing new guidelines in cervical cancer screening through the use of screen-and-treat strategies [11, 12]. In Haiti, single visit screen-and-treat approaches for cervical cancer with visual inspection with acetic acid (VIA) , followed by treatment of positive screens with cryotherapy have been implemented in certain settings [13, 14]. However these interventions heavily rely on sporadic interventions through non-governmental and civil society organizations [14–16].

Studies have evaluated individual barriers and facilitators associated with the uptake in cervical cancer screening in LMICs including low knowledge, perceived risk, competing health needs, lack of social support, and cultural beliefs [17, 18]. While many studies cite the influence of these individual-level factors as a primary motivator of health behaviors surrounding cancer screening, few studies have adequately explored the role of health literacy and sources of information in cervical cancer screening in LMICs. Furthermore, health behavior change theories such as the Health Belief Model largely overlook the construct of health literacy in engagement in preventive behaviors. Health literacy and sources of information may contribute towards individuals' cervical cancer and cervical cancer screening knowledge levels, perceived susceptibility, perceived benefits and barriers, perceived severity, and perceived self-efficacy in engagement in cervical cancer screening; thus informing perceptions and attitudes towards engagement in cancer screening [17, 19].

In the context of cancer screening, a growing body of literature has observed relationships between high literacy levels and access to information sources as determinants of engagement in cancer screening [20, 21]. High literacy and access to health information have been found to contribute to positive outcomes of knowledge, attitudes, and subsequent engagement in preventive health behaviors [23–25]. In several LMICs, studies have demonstrated that low levels

of cancer literacy can result in low uptake of screening even among populations of higher socioeconomic status [22, 26]. In relation to sources of information, health literacy has been found to be influenced by information sources, with certain populations being more prone to certain information sources compared to others which can impact engagement in screening interventions [27].

The WHO strategy for cervical cancer prevention and control emphasizes the essential role of health education in cervical cancer prevention programs, with lack of information being a deterrant to engagement in screening [21]. Research efforts to understand the influence of literacy and sources of information on cervical cancer screening among Haitian women have been largely insufficient. There is a growing need for the development of initiatives to take into consideration literacy, language barriers, and social inequities that may contribute to cervical cancer risk among Haitian women [21]. The purpose of this study was to assess determinants of engagement in cervical cancer screening among women living in Haiti, with a focus on sociodemographics, literacy, and access to sources of information.

## Methods

### Ethics statement

Data from the Haitian USAID Demographic and Health Surveys (DHS) wasiswere used for this study [28]. DHS is a nationally representative population-based survey with Women of reproductive age (15–49 years old) being eligible to participate with all data being anonymous.

Procedures and questionnaires for DHS surveys are reviewed and approved by the International Coaching Federation (ICF) International Review Board (IRB), with country-specific DHS surveys being reviewed internally by host countries [28, 29].

### Data source

The most recent Haitian DHS data was collected in 2017–2018, however for the purposes of this study DHS data from the year 2016–2017 was used as this dataset contained specific cervical cancer screening indicators. In 2016–17, MSPP in partnership with several organizations such as Partners in Health (i.e. Zanmi Lasante) led efforts to increase access to primary and secondary cervical cancer prevention services [30]. The widespread cervical cancer awareness campaigns in Haiti during this time period may have contributed to the inclusion of cervical cancer measures on the DHS survey. Authorization to use the dataset was obtained through submitting a data use agreement which was approved by ICF.

### Measures

Items from the the DHS survey were used for this analysis. The survey included items measuring sociodemographics, reproductive behavior and intentions, family planning, HIV and STI knowledge, marital background,intimate partner violence,maternal care, and other topics of public health significance.

The dependent variable in our study was engagement in cervical cancer screening. This was measured as having ever recieved cervical cancer screening, which was reported as a binary variable. The independent variables included sociodemographics, literacy levels, and access and frequency in using information sources. These variables have been associated with engagement in cervical cancer screening in previous studies within LMICs [5, 10, 12, 31, 32]. In addition, given increased cervical cancer campaigns in Haiti in 2016–17, it is possible that participants may have had exposure to cervical cancer screening information, which may have influenced their engagement in screening services.

More specifically, sociodemographics included age, number of children, marital status, area and department of residence, economic status, education level, insurance status, employment, religion, engagement in the health system (i.e. having ever received prenatal care), transportation and age of first coitus which were measured as categorical variables. Literacy level was measured as a nominal variable (e.g. can read a partial sentence, can read a full sentence, cannot read at all, blind/visually impaired) which was converted to a binary variable (partial or full literacy [i.e.able to read a partial or full sentence], no literacy [i.e. cannot read or blind/visually impaired]). Language of literacy was not asked on the DHS survey. Sources of health information included items measuring household ownership of communication devices (i.e. radio, television, newspapers, internet), along with the frequency of using communication devices. The DHS dataset did not directly measure whether participants heard of cervical cancer through information sources. However, the dataset measured whether women heard of family planning using information sources with women reporting having heard of family planning through either radio, newspaper or magazine, television, or text message. Thus, radio, TV, and print materials were selected as indicators of sources of information in our study. Such platforms are widely used in Haiti by NGOs and MSPP to disseminate health education, and have been frequently used to spread information on cervical cancer and prevention services along with other disease states [14, 33, 34].

## Data analysis

STATA software was used for the analysis [35]. Descriptive statistics were used to describe the sample. Univariate logistic regression was conducted to evaluate the significance of covariates on engagement in cervical cancer screening. Variables that demonstrated significance in the univariate regression were included in the multivariate logistic regression models. Two multivariate models were developed, the first evaluated the influence of sociodemographics on receiving cervical cancer screening. The second model evaluated the influence of literacy and sources of information on receiving cervical cancer screening. In all analyses, statistical significance was set at $p \leq 0.05$.

## Results

There were N = 6,530 women in the total sample, however n = 610 women answered questions pertaining to engaging in cervical cancer screening. Thus, the results for this study focus on the sub-set of women who responded to cervical cancer screening questions. Over seventy percent of women (n = 444; 72.79%) had heard of cervical cancer, however fewer women had heard of tests for cervical cancer (n = 295; 66.44%). Only n = 45 (7.38%) women reported ever testing for cervical cancer. Of those who tested for cervical cancer, the majority had tested within the last 1–3 years (n = 21, 46.67%).

## Sociodemographics

While the total sample included women between the ages of 15–49 years, only women between 35–49 years answered cervical cancer screening questions. In assessing sociodemographic variables, the majority of women lived in rural areas (n = 449, 73.60%). Although over half of the sample was employed (n = 402; 65.90%) , the socioeconomic status of the sample was low. Regarding education level, the majority had no education (n = 233; 38.20%). Among those who received education, primary school was the highest level completed (n = 220; 36.07%). The vast majority of women were married (n = 498; 81.64%) with children, and had a history of receiving prenatal care (n = 409; 88.34%). Over seventy percent of participants were sexually active within the last two months (n = 457;74.92%) (Table 1).

**Table 1. Sociodemographic characteristics of sample.**

| Characteristic | N (%) |
|---|---|
| **Age** | |
| 35–39 | 359 (58.85) |
| 40–44 | 192 (31.48) |
| 45–49 | 59 (9.67) |
| **Residence** | |
| Urban | 161 (26.39) |
| Rural | 449 (73.61) |
| **Economic Status** | |
| Poor-poorest | 364 (59.68) |
| Middle | 104 (17.05) |
| Rich-Richest | 142 (23.28) |
| **Education Level** | |
| No education | 233 (38.20) |
| Primary | 220 (36.07) |
| Secondary | 137 (22.50) |
| Higher | 20 (3.38) |
| **Employment Status** | |
| Unemployed | 208 (34.10) |
| Employed | 402 (65.90) |
| **Religion** | |
| No religion | 43 (7.05) |
| Catholic | 246 (40.33) |
| Protestant | 312 (51.15) |
| Voudousant | 9 (1.48) |
| **Marital Status** | |
| Married | 498 (81.64) |
| Living with partner | 53 (8.69) |
| Never in union, widowed, divorced, or separated | 59 (9.67) |
| **Number of Children** | |
| 1–3 | 152 (24.92) |
| 4–7 | 319 (52.30) |
| 8–10 | 112 (18.36) |
| >10 | 27(4.43) |
| **Having received hospital prenatal care (N = 463)** | |
| No | 54 (11.66) |
| Yes | 409 (88.34) |
| **Age at first sex** | |
| 8–12 | 19 (3.11) |
| 13–17 | 315 (51.64) |
| 18–22 | 214 (35.08) |
| 23–27 | 43 (7.05) |
| >30 | 19 (3.11) |
| **Transportation** | |
| No Transportation | 523 (85.73) |
| Access to Transportation | 77 (12.62) |
| **Department** | |
| Aire Metropolitaine/Ouest | 132 (21.64) |

(*Continued*)

**Table 1.** (Continued)

| Characteristic | N (%) |
|---|---|
| Sud-est | 35 (5.74) |
| Nord | 48 (7.87) |
| Nord-est | 51 (8.36) |
| Artibonite | 92 (15.08) |
| Centre | 54 (8.85) |
| Sud | 62 (10.16) |
| Grand-Anse | 48 (7.87) |
| Nord-ouest | 61 (10.00) |
| Nippes | 27 (4.43) |

The univariate regression model showed that total children, economic status, education level, health insurance, religion, and access to transportation were all associated with a statistically significant higher odds of reciving cervical cancer screening. Using the significant sociodemographic variables from the univariate model, a multivariate model was developed (Table 2). In the multivariate model, women from an economic background of poorer (aOR = 4.06, 95% CI [1.16,14.27], p = 0.03) and richest (aOR = 19.10 , 95% CI[2.58,141.57], p = 0.00), women with the highest education level (aOR 7.58 , 95% CI [1.64,34.97], p = 0.00) and those with insurance (aOR = 16.40, [95% CI 2.65, 101.42], p = 0.00) had a statistically significant higher odds of undergoing cervical cancer screening.

## Literacy and sources of information

The sample had nearly equal numbers of women who were illiterate (n = 303; 49.67%) and literate (n = 307; 50.33%). Among those who were literate, 10.75% (n = 33) had been screened. Among those who were not literate , less than 5% (3.96%, n = 12) had received screening. In comparing access to information sources, a higher number of participants owned mobile phones (n = 219, 31.90%). Regarding frequency of usage of sources of information, radios were more frequently used with the majority of participants reporting listening to radio between at least once a week (n = 132 ; 21.64) to almost every day (n = 174;28.52%) (Table 3).

All the co-variates for sources of information demonstrated significance in the univariate analysis. In the multivariate model, women who had access to a television (aOR = 4.28, 95% CI [1.21,9.34], p = 0.02) and those who owned a mobile phone (aOR = 4.44, 95% CI [1.00,5.59], p = 0.05) were over four times more likely to undergo cervical cancer screening, with this finding being statistically significant. In the multivariate model , women who read the newspaper at least once per week were over three times more likely to engage in cervical cancer screening (aOR = 3.57, [95% CI 1.10,11.59] p = 0.03). Women who had some form of internet use within the last 12 months were four times more likely to undergo cervical cancer screening (aOR = 3.99 [95% CI 1.45,10.97] p = 0.01) (Table 4).

## Discussion

This study set out to assess the role of sociodemographics, health literacy, and sources of information on cervical cancer screening in Haiti. In Haiti, cervical cancer is a disease of major public health significance. Estimates from GLOBOCAN, a database developed by the International Agency for Research on Cancer (IARC), rank the Latin America and the Caribbean regions generally with some of the highest cervical cancer incidence internationally, and Haiti specifically with disproportionate incidence [4, 36, 37]. Despite these alarming data, reports

**Table 2. Regression analysis of sociodemographic variables on cervical cancer screening.**

| Covariates | OR [95% CI] | P Value | AOR [95%CI] | P Value |
|---|---|---|---|---|
| **Economic Status** | | | | |
| Poorest | *REF | | *REF | |
| Poorer | 4.44 [1.36,14.47] | 0.01 | 4.06 [1.16,14.27] | 0.03 |
| Middle | 3.43 [0.95,12.42] | 0.06 | 2.82 [0.62,12.90] | 0.18 |
| Richer | 5.89 [1.73,20.13] | 0.01 | 3.81 [0.58,24.89] | 0.16 |
| Richest | 23.22 [7.43,72.52] | 0.00 | 19.10 [2.58,141.57] | 0.00 |
| **Education Level** | | | | |
| No education | *REF | | *REF | |
| Primary | 1.33 [0.52,3.46] | 0.55 | 0.90 [0.32,2.52] | 0.84 |
| Secondary | 3.98 [1.67,9.50] | 0.00 | 1.39 [0.45,4.26] | 0.56 |
| Higher | 28.13 [9.13,86.63] | 0.00 | 7.58 [1.64,34.97] | 0.01 |
| **Insurance Status** | | | | |
| Not Insured | *REF | | *REF | |
| Insured | 34.51 [8.58,138.80] | 0.00 | 16.40 [2.65,101.42] | 0.00 |
| **Transportation** | | | | |
| No transportation | *REF | | *REF | |
| Access to Transportation | 2.22 [1.04,4.70] | 0.04 | 0.44 [0.13,1.48] | 0.12 |
| **Residence** | | | | |
| Urban | *REF | | *REF | |
| Rural | 0.23 [0.12,0.43] | 0.00 | 1.44 [0.39,5.29] | 0.58 |
| **Department** | | | | |
| Aire Metropolitaine/Ouest | *REF | | *REF | |
| Sud-est | 0.11 [0.01,0.85] | 0.04 | 0.17 [0.01,2.07] | 0.16 |
| Nord | 0.16 [0.03,0.73] | 0.02 | 0.14 [0.0,1.18] | 0.07 |
| Nord-est | 0.40 [0.13,1.18] | 0.10 | 1.57 [0.37,6.62] | 0.54 |
| Artibonite | 0.17 [0.05,0.53] | 0.00 | 0.42 [0.09,1.82] | 0.24 |
| Centre | 0.37 [0.13,1.10] | 0.08 | 1.38 [0.31,6.09] | 0.67 |
| Sud | 0.62 [0.25,1.54] | 0.30 | 1.85 [0.45,7.69] | 0.40 |
| Grand ' anse | 0.16 [0.03,0.73] | 0.02 | 0.59 [0.09,3.91] | 0.59 |
| Nord-ouest | 0.06 [0.01,0.48] | 0.01 | 0.21 [0.02,2.00] | 0.12 |
| Nippes | 0.14 [0.02,1.12] | 0.07 | 0.46 [0.04,5.00] | 0.52 |

have shown that Haitian women's participation in cervical cancer screening is drastically low due to structural and individual barriers [9, 10] Our study aimed to address the individual barriers that may contribute to low screening uptake with a focus on health literacy and sources of information as this has been shown to contribute to cervical cancer screening uptake. Major contributing factors to low uptake in our sample, included disadvantageous sociodemographic status and restricted access to sources of information which may limit awareness and knowledge towards cervical cancer and preventative services.

The first set of models evaluated the association of sociodemographics on receiving cervical cancer screening. While univariate analysis resulted in significance for engagement in cervical cancer screening across total children, economic status, education level, health insurance, religion, and transportation, the multivariate model indicated that women from extremes of economic status (i.e., poorer and richest) were most likely to have engaged in screening. This initial finding may suggest not only the protective nature of high education often associated with better access to healthcare resources, but also a potential growing impact of health interventions focused on underresourced populations [5, 38, 39]. Consistent with previous studies,

**Table 3. Literacy and sources of information characteristics.**

| Characteristic | N(%) |
|---|---|
| **Literacy** | |
| Cannot read | 303 (49.67) |
| Partial or full literacy | 307(50.33) |
| **Household sources of information** | |
| *Household has a radio* | |
| No | 401 (65.74) |
| Yes | 199 (32.62) |
| *Household has a tv* | |
| No | 494 (80.98) |
| Yes | 106 (17.38) |
| *Mobile phone* | |
| No | 391 (64.10) |
| Yes | 219 (35.90) |
| **Frequency of using sources of information** | |
| *Reading Newspaper* | |
| Not at all | 481 (78.85) |
| Less than once a week | 81 (13.28) |
| At least once a week | 25 (4.10) |
| Almost every day | 23 (3.77) |
| *Listening to radio* | |
| Not at all | 96 (15.74) |
| Less than onceonce a week | 208 (34.10) |
| At least once a week | 132 (21.64) |
| Almost every day | 174(28.52) |
| *Watching TV* | |
| Not at all | 368 (60.33) |
| Less than onceonce a week | 164 (26.89) |
| At least once a week | 40 (6.56) |
| Almost every day | 38 (6.23) |
| *Internet Use* | |
| Never | 547 (89.67) |
| Yes, within the last 12 months | 219 (54) |
| Yes, before the last 12 months | 9 (1.48) |

women with the highest education level as well as those with insurance also had a significantly higher odds of undergoing cervical cancer screening [10, 40, 41]. However, the wide confidence interval for insurance suggests the sample does not provide precise representation of population given that majority of Haitians do not have insurance, and there is currently no national insurance program available [42]. Concerningly, only individuals between the ages of 35–49 answered questions pertaining to cervical cancer screening. DeGennaro et al. (2019) reported that in a cohort of Haitian women undergoing chemotherapy for cervical cancer, a large percentage of women were younger than 50 years old [14]. It is likely that women in Haiti have a higher chance of developing cervical cancer at younger ages due to numerous risk factors, thus reinforcing the need for regular screening starting from 25 years old as recommended by the WHO [43].

The second set of models evaluated the influence of sources of information and literacy on receiving cervical cancer screening. Notably, the sample was fairly split between literate and

**Table 4. Regression analysis of literacy and sources of information variables (Model 2).**

| Covariates | OR [95%CI] | P Value | AOR [95% CI] | P Value |
|---|---|---|---|---|
| **Literacy Level** | | | | |
| Cannot read | *REF | | *REF | |
| Partial or full literacy | 2.92 [1.47,5.77] | 0.00 | 0.77 [0.29,2.03] | 0.60 |
| **Household sources of information** | | | | |
| *Household has a radio* | | | | |
| No | *REF | | *REF | |
| Yes | 2.03 [1.09,3.79] | 0.03 | 0.47 [0.18,1.23] | 0.12 |
| *Household has a tv* | | | | |
| No | *REF | | *REF | |
| Yes | 4.28 [2.25,8.14] | 0.00 | 3.37 [1.21,9.34] | 0.02 |
| *Mobile phone ownership* | | | | |
| No | *REF | | *REF | |
| Yes | 4.44 [2.31,8.55] | 0.00 | 2.37 [1.00,5.59] | 0.05 |
| **Frequency of using sources of information** | | | | |
| *Reading Newspaper* | | | | |
| Not at all | *REF | | *REF | |
| Less than once a week | 2.28 [1.06,5.32] | 0.04 | 0.84 [0.30,2.40] | 0.75 |
| At least once a week | 7.40 [2.82,19.43] | 0.00 | 3.57 [1.10,11.59] | 0.03 |
| Almost every day | 5.29 [1.81,15.46] | 0.00 | 2.01 [0.55,7.31] | 0.29 |
| *Listening to Radio* | | | | |
| Not at all | *REF | | *REF | |
| Less than once a week | 1.16 [0.35,3.80] | 0.80 | 1.00 [0.27,3.62] | 1.00 |
| At least once a week | 1.29 [0.37,4.53] | 0.69 | 1.54 [0.40,5.87] | 0.53 |
| Almost every day | 3.68 [1.23,10.94] | 0.02 | 2.49 [0.66,9.41] | 0.18 |
| *Watching TV* | | | | |
| Not at all | *REF | | *REF | |
| Less than once a week | 1.08 [0.51,2.27] | 0.84 | 0.57 [0.22,1.46] | 0.24 |
| At least once a week | 1.22 [0.35,4.24] | 0.78 | 0.19 [0.04,0.93] | 0.04 |
| Almost every day | 4 [1.65,9.71] | 0.00 | 0.48 [0.13,1.76] | 0.27 |
| *Internet Use* | | | | |
| Never | *REF | | *REF | |
| Yes, within the last 12 months | 7.71 [3.77,15.74] | 0.00 | 3.99 [1.45,10.97] | 0.01 |
| Yes, before the last 12 months | 16.04 [4.06,63.24] | 0.00 | 13.70 [2.52,74.51] | 0.00 |

non-literate women. Nonetheless, approximately a third of the sample owned a mobile phone and nearly a quarter of the sample listened to the radio daily. Although all univariate analyses demonstrated significance, multivariate results specifically indicated women who owned a mobile phone or had some form of internet use in the last year were both four-fold more likely to undergo cervical cancer screening. This may be influenced by sociodemographics such as education and economic status, but also may suggest potential avenues of mobile Health (mHealth) for education and intervention given the ability to overcome literacy issues using audiovisual communication [44–47]. Furthermore, findings imply access to technology may be a protective factor for health care access since there was no significant difference in likelihood of undergoing cervical cancer screening by reports of those who did and did not watch television. However, those with access to a television were more likely to have undergone screening.

Studies have consistently demonstrated links between health literacy and engagement in cervical cancer screening , with low health literacy being associated with low cervical cancer knowledge and subsequent poor uptake in cervical cancer screening [7, 48, 49]. Interestingly, in our study, women who read the newspaper at least once per week were over three times more likely to engage in cervical cancer screening [40]. Yet in our analysis, literacy was not a significant predictor of cervical cancer screening. It is critical to emphasize that in Haiti, socio-cultural contexts are highly relevant when evaluating the relationship between literacy and uptake of health promotion behaviors. Haiti is one of the few countries where although one language is spoken by all citizens (e.g. Haitian Creole), the educational system uses French as the primary language of communication [50]. Educational and print materials in Haiti are often developed in French as opposed to Haitian Creole , despite less than 10% of the population speaking and reading French fluently [51, 52]. The DHS survey items measuring literacy did not specify whether participants were literate in French or Creole. It is likely that the majority of the individuals who were literate in our sample, were primarily literate in Haitian Creole. It is also likely that those who read the newspaper frequently were fluent in French given that French literacy is required to read the majority of Haitian newspapers. This may explain the reasoning as to why literacy was not significant in our analysis, yet reading the newspaper weekly was significant. French fluency may result in higher exposure to health information leading to increased engagement in health behaviors such as cervical cancer screening. Accurately exploring the relationship between literacy and health behaviors in Haiti requires researchers to account for these contextual subtleties which were not measured in this dataset. While numerous studies provide links between the influence of broad indicators of social determinants of health on linkage to cervical cancer screening and treatment, our study is one of the first to provide preliminary insight on how linguistic preferences rooted in colonialism and classism may influence engagement in cervical cancer screening in Haiti. Moise et al. (2021) [3] states that Haitian women's health should be contextualized through a syndemic approach using both a biomedical and anthropological lens; as the French language disadvantages individuals of low socioeconomic status who are primarily Creole speaking. These specific nuances are imperative to understand when working with Haitian communities, and provides significance to the importance of ensuring that research with Haitians are led by individuals who are aware of these critical contexts.

## Limitations

This study is not without limitations. Less than 10% of the sample from the DHS dataset answered questions pertaining to cervical cancer screening. In conducting the DHS survey, MSPP specifically targeted women 35–64 years old to answer questions pertaining to cervical cancer screening [6]. This is a limitation as insights from younger women who are sexually active were not obtained. Given that this is secondary data analysis, it is not possible to distinguish whether bias in data collection was present. In addition, most participants in the study were from rural and critically underserved parts of Haiti. As such, the data obtained could be skewed as most of it is obtained from participants in areas of the country with limited access to health infrastructure. It is possible there are other indicators that influenced engagement in cervical cancer screening that were not included in this analysis.

## Conclusion and future research

This paper adds to the literature by providing associations between sociodemographics, literacy, and sources of information on cervical cancer screening among women in Haiti. Future research should focus on recruiting from a larger sample, and including more participants

from metropolitan areas, to evaluate whether the uptake of cervical cancer screening remains low regardless of social or demographic status. Future research should also evaluate how health communication efforts in Haiti can be tailored to enhance women's knowledge of HPV and cervical cancer, which can lead to higher rates of cervical cancer screening; particularly in light of contextual nuances pertaining to literacy [53, 54]. Best-approaches in knowledge dissemination (e.g. marketing campaigns, expansion of audiovisual media) with the goal of influencing motivation and intention to screen should be assessed [55, 56]. For instance, the use of social media and mobile phone platforms that provide interactive audiovisual content in both Haitian Creole and French, may improve motivation in engaging in cervical cancer prevention for communities at-large. Lastly, more implementation research is needed in expanding cervical cancer prevention efforts in Haiti, and evaluating how key stakeholders can influence the scale-up of cervical cancer prevention efforts in a setting with numerous health challenges [57]. Of note, the incidence of high-risk strains of HPV are likely to increase among Haitan womeni due to the current sociopolitical crisis which has resulted in unforseen rates of sexual violence [58]. Therefore, ensuring women have access to cervical cancer screening including timely notification of positive results, along with additional sexual and reproductive health services should be a public health priority.

## Author Contributions

**Conceptualization:** Dominique Guillaume.

**Data curation:** Dominique Guillaume.

**Formal analysis:** Dominique Guillaume, Claire Rolland.

**Funding acquisition:** Dominique Guillaume.

**Investigation:** Dominique Guillaume.

**Methodology:** Dominique Guillaume.

**Project administration:** Dominique Guillaume.

**Supervision:** Dominique Guillaume, Natalie Pierre-Joseph.

**Validation:** Rhoda Moise.

**Writing – original draft:** Dominique Guillaume, Rhoda Moise.

**Writing – review & editing:** Dominique Guillaume, Rhoda Moise, Joyline Chepkorir, Kamila Alexander, Maria Luisa Alcaide, Rasheeta Chandler, Natalie Pierre-Joseph.

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
