## [Decision Letter · Decision Letter 0]

5 Jun 2023

PGPH-D-23-00669

Links between sociodemographics, literacy, and sources of information on cervical cancer screening in Haiti: A secondary data analysis of 2016-17 DHS surveys

Dear Dr. Guillaume-Rolland,

Thank you for submitting your manuscript to PLOS Global Public Health. After careful consideration, we feel that it has merit but does not fully meet PLOS Global Public Health’s publication criteria as it currently stands. Therefore, we invite you to submit a revised version of the manuscript that addresses the points raised during the review process.

EDITOR: 

Kindly consider all revisions recommended by the reviewers, especially, typographical issues,Review the title of the study and consider the justifications offered for such modifications, Remember that this study is a scientific research and is focused on modifiable risk factors involving risk behaviour among women and the theoretical basis of the context should feature strongly as the foundation of any elucidations the study offers.

We look forward to receiving your revised manuscript.

Kind regards,

Nnodimele Onuigbo Atulomah, PhD

Academic Editor

Journal Requirements:

Additional Editor Comments:

The subject of this study is pertinent at this time considering the burden placed on the health system of the country as identified. I have made additional observations the authors should seriously consider in revising the manuscript to significantly improve what it is to offer the audience intended. Kindly consider what the reviewers have suggested.

TITLE: There is need to revise the title of the study to read; "Sociodemographics and Health-literacy as predictors of cervical cancer screening Practices among Haitian women: A secondary data analysis of 2016-17 DHS surveys". The rationale is that the use of Health-literacy best contextualize the argument of the study as mentioned in the discussion and many instances literacy and information sources in the context understanding the issues around cancer of the cervix are modifiable risk factors that can be ameliorated by health education through the identified information media.

ABSTRACT:

(1) The media through which the at-risk population may access health-related information may be regarded as important resources to improve health literacy. Therefore revise the objective to align with health-literacy. Kindly note that the conclusion also justifies the basis of replacing literacy and sources of information with health-Literacy as a construct in the study.

INTRODUCTION:

In the introduction, the second sentence should read: "Haiti reports the highest rates of cervical cancer incidence and mortality…"

Second paragraph argues of favorable preventable outcomes related to public health principles of prevention and control but fail to adequately review the scientific context necessary to elucidate the dynamics of the observed poor screening uptake the basis of which constitute the modifiable risk factors the variables in the study are predicated. It would have been very appropriate to review the theoretical clarifications needed to understand the dynamics attempted in the third paragraph: "In the context of cancer screening, a growing body….may contribute to cervical cancer risk among Haitian women"

The omission of an articulated underpinning theoretical foundation involving the modifiable risk factors in this study representing the framework constitute a serious omissions that invalidates any attempt to elucidate the dynamics of the observed outcomes of poor screening practices. The argument appears superficial and not grounded in health promotion theories, though mention is made about "primary prevention", "secondary prevention" "utilization of screening" these are not linked with the source theories and cannot provide the needed proof of concept the objective of the study seeks to elucidate.

The mention of "preventive health behaviour" and all the constructs within the context of health-seeking governing preventive health explainable by the health belief model demands the review of this foundational framework before the statement of purpose "...to assess determinants of engagement in cervical cancer screening among…".

RESULTS:(Subsection reporting Literacy and Sources of Information)

There is a need to revise the statement in the first sentence: "...sample had nearly equal amounts of women who..." to "...sample had nearly equal numbers of women who..."

In the results how did they compare with screening outcomes?

DISCUSSION:

In initiating the discussion for this study, it would have been very appropriate to reiterate the objectives of the study to maintain alignment with the thesis of the study. The thesis of the study includes emerging questions warranting the study needing answers such as why is cervical cancer highest in Haiti, why are Haitian women not engaged in cervical screening practices for likely early detection or even preventive vaccination practices among others? Then proceed to review finding and explain likely reasons for what the data has revealed. The would strengthen such a fine scientific work.

The second paragraph begins with a statement; "...evaluated the influence of sociodemographics on receiving cervical cancer screening." It would be most appropriate to use "associated with" rather than "influence" this study cannot demonstrate influence but association or demonstrate predictive value of variables. Only experimental studies can infer impact, influence or effect.

Were there measures of engagement in screening in the study?

Reviewers' comments:

Reviewer's Responses to Questions

**Comments to the Author**

1. Does this manuscript meet PLOS Global Public Health’s publication criteria? Is the manuscript technically sound, and do the data support the conclusions? The manuscript must describe methodologically and ethically rigorous research with conclusions that are appropriately drawn based on the data presented.

Reviewer #1: Yes

Reviewer #2: Yes

2. Has the statistical analysis been performed appropriately and rigorously?

Reviewer #1: Yes

Reviewer #2: Yes

3. Have the authors made all data underlying the findings in their manuscript fully available (please refer to the Data Availability Statement at the start of the manuscript PDF file)?

Reviewer #1: Yes

Reviewer #2: No

4. Is the manuscript presented in an intelligible fashion and written in standard English?

Reviewer #1: Yes

Reviewer #2: Yes

5. Review Comments to the Author

Reviewer #1: Major Issues

• The authors need to provide the following data on Age, Religion, Marital Status and Employment Status and insert into Table 2. This is because the authors have mentioned that these socio-demographic data were part of what was analysed. These variables are missing in the Table 2. Hence it will be wrong to deduce that these are statistically significant with the odds of undergoing cervical screening.

• Studies have reported that these variables Age, Religion, Marital Status and Employment Status are significant to determine whether or not women will undergo cervical screening. See the following Articles Below as a reference.

• e study.

Minor Issues

• Correct “Ghe” to “The” on Line 7 under Data Analysis section of the Paper.

• The authors should incorporate the word “variables” to buttress their explanation of the socio-demographic data set.

Reviewer #2: The manuscript is technically sound, and the data supports the conclusions. The manuscript describes methodologically and ethically rigorous research with conclusions that are appropriately drawn based on the data presented.

The authors did not make all data underlying the findings in their manuscript fully available. This is most likely due to the fact that third party data - from the Haitian USAID Demographic and Health Surveys (DHS) - was analyzed in the study.

The language of the manuscript is clear, correct, and unambiguous. However some typographical and grammatical errors were identified in the attached manuscript.

6. PLOS authors have the option to publish the peer review history of their article (what does this mean?). If published, this will include your full peer review and any attached files.

**Do you want your identity to be public for this peer review?** For information about this choice, including consent withdrawal, please see our Privacy Policy.

Reviewer #1: **Yes: **Saheed Akinmayowa Lawal

Reviewer #2: **Yes: **Ayodeji O. OLARINMOYE

---

## [Editor Report · Decision Letter 1]

11 Jul 2023

Sociodemographics and health-literacy as predictors of cervical cancer screening practices among Haitian women: A secondary data analysis of 2016-17 DHS surveys

PGPH-D-23-00669R1

Dear Dr. Guillaume-Rolland

We are pleased to inform you that your manuscript 'Sociodemographics and health-literacy as predictors of cervical cancer screening practices among Haitian women: A secondary data analysis of 2016-17 DHS surveys' has been provisionally accepted for publication in PLOS Global Public Health.

Best regards,

Nnodimele Onuigbo Atulomah, PhD

Academic Editor

Congratulations for your resilience in following through with all recommended revisions.